# Enhancing Multi-Agent Learning in Real-World Interactive Environments through Process Reward Decomposition

## Abstract

LLM-based agents have made significant advancements in interactive environments, such as mobile operations and web browsing, with multi-agent systems further boosting performance. However, current agent learning techniques heavily rely on in-domain data and struggle to generalize across tasks and environments. Moreover, existing multi-agent learning methods are limited by fixed role assignments, which restrict their flexibility and generalization. Furthermore, the multi-step nature of interactive tasks, combined with sparse end-to-end reward signals, hinder effective learning to a great extent. To address these issues, we propose *CollabUIAgents*, a two-stage multi-agent learning framework for interactive environments. In the first stage, the base model is adapted to the environment using curriculum learning on multi-level instruction data. In the second stage, a novel process reward decomposition strategy is introduced during reinforcement learning, allowing rewards to be distributed at both the agent and conversation round levels. This granular feedback fosters collaborative awareness among agents without predefined roles and improves learning efficacy. Experimental results show that our method significantly enhances the performance of multi-agent systems based on open-source models, achieving notable improvements both within and across domains, while also exhibiting strong cross-environment generalization capabilities. Moreover, our best-performing systems achieve results on par with or exceed those of the strong closed-source models, while maintaining the flexibility to be integrated with prompt-based multi-agent systems for future research.

## 1 Introduction

Autonomous agents have made substantial progress in interactive environments, such as mobile operations and web browsing, by leveraging large language models (LLMs). These agents hold immense potential not only to automate repetitive tasks but also to enhance decision-making and streamline complex workflows. As a result, they can free up human resources for higher-level problem-solving and innovation. The increasing interest in developing such agents is evident in the growing body of work on, for instance, mobile environment simulators (Rawles et al., 2024; 2023; Zhang et al., 2024c; Deng et al., 2024a; Wang et al., 2024c), web browsing benchmarks (Shi et al., 2017; Liu et al., 2018a; Yao et al., 2022a; Zhou et al., 2024b; Deng et al., 2023; 2024b), and LLM-based agents targeting on mobile and web tasks, including single-agent (Yan et al., 2023; Lai et al., 2024; Bishop et al., 2024; Wang et al., 2024b; Hong et al., 2024; Cheng et al., 2024) and multi-agent systems (Wang et al., 2024a; Zhou et al., 2023; Liu et al., 2024; Zhang et al., 2024d).

However, current efforts in LLM-based agent learning still face several challenges in these kind of interactive environments. (1) Single-agent learning methods (Chen et al., 2023a; Gur et al., 2024; Furuta et al., 2024) heavily relies on in-domain data (e.g., HTML-formatted inputs), which restricts its ability to generalize across diverse tasks and environments, such as transitioning between web environments using HTML and mobile environments using Android automator. Despite being trained on vast amounts of data from diverse domains, single agent based on open-source LLMs (Zeng et al., 2023; Zhang et al., 2024b) demonstrate only moderate generalization capabilities and continue to lag behind closed-source models. (2) Although multi-agent learning methods (Qiao et al., 2024; Liang et al., 2024) have better performance, they are often constrained by rigid role assignments, which

limits their adaptability to unseen environments. For instance, an agent designed to retrieve documents for question answering may struggle to handle file operations in a mobile environment. (3) In addition, multi-step nature of interactive tasks results in sparse reward signals during end-to-end learning, which complicates effective learning in real-world interactive environments.

In this work, we introduce a two-stage multi-agent learning framework, named *CollabUIAgents*, designed to address challenges in real-world interactive environments. The framework is structured without predefined roles in the multi-agent system or domain-specific data collection requirements. Specially, stage 1 focuses on enabling the base model to adapt to the environment through curriculum learning on multi-level instruction data, aimed at **learning general environmental knowledge**. To facilitate this process, we propose a fully automated data synthesis strategy that significantly reduces labor costs and accelerates data acquisition. The synthesized instruction data comprises three parts: (1) basic environmental knowledge, (2) simple instruction knowledge, and (3) process preference knowledge, with a progressively increasing level of difficulty. The base model is first fine-tuned using Supervised Fine-Tuning (SFT) (Ouyang et al., 2024) on the first two data segments, followed by Direct Preference Optimization (DPO) (Rafailov et al., 2024) using the process preference data. Stage 2 introduces a novel process reward decomposition strategy within the framework of **multi-agent reinforcement learning (MARL)**, allocating rewards at both the agent and conversation round levels. Similar to the preference data synthesis in stage 1, the preference data in this stage are labeled with fine-grained reward signals by a multi-agent data synthesis pipeline. Instead of assigning a single reward label at each step, the pipeline assesses the contributions of each agent during each conversation round and allocates rewards accordingly, which is known as *process reward* (Uesato et al., 2022). This approach enables a VDPPO-style (Ma & Luo, 2022) training process, fostering collaborative awareness among the agents.

Our framework provides much more granular feedback on each agent's contribution throughout the task, enhancing learning effectiveness over previous works. And this framework is also capable of cross-environment user interface (UI) interaction, supporting both mobile and web environments, either through directly applying multi-agent systems adapted from mobile environments to websites or through continue MARL on the new environment.

Experimental results demonstrate that the proposed multi-agent system achieves superior performance compared to existing methods, including surpassing the strong closed-source model Gemini 1.5 Pro (Gemini Team Google, 2024) and achieving performance comparable to GPT-4 (OpenAI, 2024) with Qwen2-7B (Yang et al., 2024) as the base model, on both in-domain and out-of-domain mobile environments. Surprisingly, CollabUIAgents demonstrates effective cross-environment generalization from mobile to web environments, under both scenarios of direct application and continue training. And the system of the latter setting also achieves comparable performance to GPT-4.

In summary, our contributions are as follows:

- We propose a two-stage multi-agent learning framework consists of general environmental knowledge learning and multi-agent reinforcement learning, named *CollabUIAgents*, which requires no human intervention in data synthesis and optimization process.
- Our method incorporate a novel process reward decomposition strategy in multi-agent reinforcement learning, providing much finer-grained reward signals on both agent and conversation levels, overcoming signal scarcity in end-to-end learning for interactive environments.
- Extensive experiments show that our proposed CollabUIAgents surpasses the performance of Gemini 1.5 Pro and shows competitiveness comparable to GPT-4 on both in-domain, out-of-domain mobile environments, and even cross-environment tasks.

## 2 METHODOLOGY

This section details the proposed *CollabUIAgents* framework, which addresses the challenges in multi-agent learning for real-world interactive environments. The methodology consists of four key components: (1) the task formulation, where we formally define the problem of applying multi-agent systems on real-world interactive environments; (2) the architecture of the CollabUIAgents framework, outlining the overall multi-agent system and agent conversations design; (3) the two-stage learning process, where agents first acquire general environmental knowledge and then optimize

their behaviors using Multi-Agent Reinforcement Learning (MARL) enhanced by Process Reward Decomposition; and (4) the cross-environment adaptation, where we describe how a multi-agent system trained in one environment can adapt and generalize to different environments.

## 2.1 FORMULATION AND NOTATION

We treat real-world interaction tasks as a sequential decision-making process with either single agent or multi-agent systems in dynamic environments. The task involves agents making decisions based on the current environment state and their accumulated interaction history.

**Task Formulation**    Let $S$ be the set of all possible states of a given interactive environment, where each $s \in S$ represents a specific configuration of the UI and hidden states at a given time step, including an initial state $s_0$ and a terminal state. The set of all possible actions that a given agent system $\mathcal{G}$ can take is denoted as $\mathcal{A}$, where $a \in \mathcal{A}$ includes actions such as clicking buttons, typing, or scrolling through content. The environment evolves according to a transition function $T$:

$$s_{t+1} = T(s_t, a_t), s_t, s_{t+1} \in S, a_t \in \mathcal{A}, \tag{1}$$

where $s_t$ is the state at time step $t$, and $a_t$ is the action taken by the agent system at that step. The task ends when reaching a terminal state or exceeding the maximum step $T_{\max}$. From the state $s_t$, the observation $o_t$ is derived as formatted description in language. Each agent $\pi_i$ in the system selects actions based on current observation $o_t$, the history of past interactions $H_{t-1} = (s_0, a_0, ..., s_{t-1}, a_{t-1})$, and the message for agent $\pi_i$ at conversation round $j$, denoted as $\mathcal{C}_t^{i,j}$, since multi-round conversations may happen at each decision step. $\mathcal{C}_t^{i,j}$ is omitted for single agents:

$$a_t^{i,j} = \pi_i\left(o_t, H_{t-1}, C_t^{i,j}\right), a_t^{i,j} \in \mathcal{A}, i = 1, ..., |\mathcal{G}|, \tag{2}$$

where $|\mathcal{G}|$ is the number agents in the system. And $a_t$ is determined by an aggregation function $f_{\text{agg}}$ (which is identity for single agents ($|\mathcal{G}| = 1$)):

$$a_t = f_{\text{agg}}\left(\left\{a_t^{i,j} \middle| i = 1, \cdots, |\mathcal{G}|; j = 1, \cdots, m\right\}\right), \tag{3}$$

where $m$ is the number of conversion rounds. The goal of the task is to maximize the reward at the terminal state over a sequence of interactions.

**Real-World Interactive Environment**    The observation and action space in real-world interactive environment are rich. Specifically, for the **mobile operation environments**, which offer an interface that allows agents to receive observations and perform actions on mobile devices, the observation space may include high-resolution screenshots and a UI tree from Android automater. The action space mirrors human interactions, featuring gestures (such as tapping, long-pressing, and swiping), typing, and navigation buttons (e.g., home and back). Complete actions are listed in Table 5. For **web browsing environments**, the observation space may include task description, simplified HTML, and current location. The HTML offers the model both structural and content details of the page, while the current location information allows it to understand its position on the webpage. Consistent with previous work, we use a unified web browsing action space in both of the aforementioned environments. The actions include hover, select, click, etc. More actions can be found in Table 6.

**Reward Function and Objective**    The reward $R_{\text{total}} \in \{0, 1\}$ is defined in the environment based on task requirements. The overall objective is to maximize the expected reward. Rewards are sparse, as only the terminal state gives out reward signals, posing a challenge to end-to-end approaches.

## 2.2 COLLABUIAGENTS FRAMEWORK

The *CollabUIAgents* framework is designed to address the issues of sparse rewards and fixed roles in multi-agent learning. It operates without predefined roles, providing fine-grained rewards, and supports generalization across different environments. The framework is composed of two main stages: General Environmental Knowledge Learning and Multi-Agent Reinforcement Learning.

### 2.2.1 MULTI-AGENT SYSTEM ARCHITECTURE

The architecture of the multi-agent system ($\mathcal{G}$) in CollabUIAgents is in consistency with previous works (Zhuge et al., 2024a; Liu et al., 2024), which consists of $|\mathcal{G}| = n$ agents, each represented

by a policy $\pi_i$ that communicate with each other through a message network $\mathcal{E}_{\mathcal{G}}$. As shown in Figure 2, the network is a directed acyclic graph (DAG), where messages are passed from $\pi_{i_1}$ to $\pi_{i_2}$ if there is an edge pointing from $\pi_{i_1}$ to $\pi_{i_2}$. Specifically, the message is from the output of $\pi_{i_1}$. It is worth noting that the architecture remains the compatibility for prompt-based agent methods, whose performance is left for future investigation. We instead use naive prompting for fair comparisons.

The agents operate in a topological order, and starting from the source to the sink node, allowing each agent to aggregate all responses from its predecessors to form $C^{i,j}$ in equation 2. We define the round of conversation as $m$. In each conversation round, all agents operate once along the topological order, and each agent could receive its own decision from the last round besides decisions from predecessors, i.e., we keep a local memory with size equal to 1. The proper size of local memory enhances the diversity of decision making and avoids introducing too long contexts. According to equation 2, at the time step $t$ to interact with the environment, the system produce an **action matrix**:

$$\boldsymbol{A}_t = (a_t^{i,j}), i = 1, ..., n; j = 1, ..., m, \tag{4}$$

where $a_t^{i,j}$ is the intermediate decision from the $i$-th agent at $j$-th conversation round for interaction step $t$, as shown in Figure 2. Then, majority voting is used to decide the final action at the time step,

$$a_t = f_{\text{agg}}(\boldsymbol{A}_t) = \operatorname{argmax}_a \sum_{i=1}^{n} \sum_{j=1}^{m} \mathbf{1}_{a_t^{i,j}=a}, \tag{5}$$

where $\mathbf{1}_{\text{condition}}$ is the indicator function. The agents are all required to output an action and collaborate towards a common objective to enlarge the expected end-to-end reward $R$, which allows them to function with the same base model for better efficiency, and operate heterogeneously due to different conversation messages.

### 2.2.2 STAGE 1: GENERAL ENVIRONMENTAL KNOWLEDGE LEARNING

The first stage of the CollabUIA-gents framework focuses on adapting agents to new environments through curriculum-based single-agent training (Bengio et al., 2009). The training data is synthesized automatically with a multi-agent data synthesis pipeline and consists of progressively complex instruction sets in three levels, designed to help agents build a strong foundation of environmental knowledge. The UI agent generate responses to synthesized queries faithfully, the adversarial agent generates negative samples, and the critic agent grades process rewards.

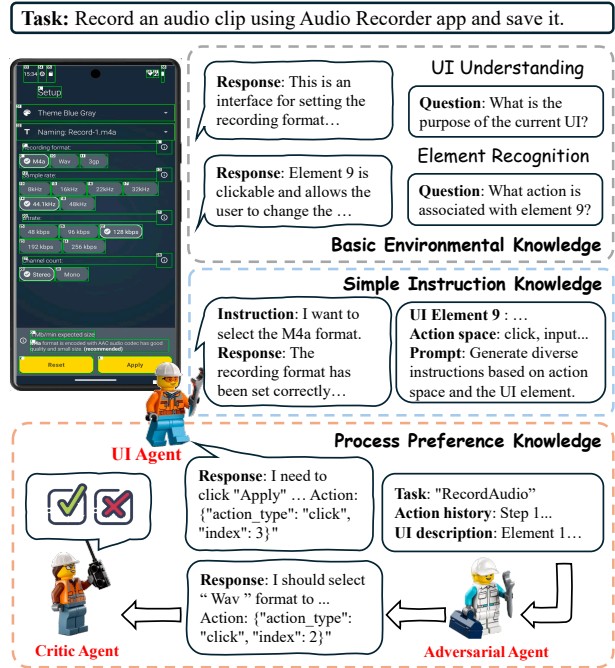

Figure 1: Our multi-agent autonomous data synthesis pipeline. Given a task, the pipeline can autonomously collect data covering basic environmental knowledge, simple instruction knowledge, and process preference knowledge in real-world interactive environments.

**Curriculum Structure** The training data is divided into three categories, as collected in Figure 1:

(1) **Basic Environmental Knowledge**: This data segment includes identifying UI elements and understanding their properties. We categorize basic knowledge into two types: **UI Understanding** (coarse-grained): This refers to a broad understanding of the layout and information contained in the UI, such as identifying the purpose of the interface. **UI Element Recognition** (fine-grained): Since UI typically contains a large number of densely packed interface, the agent needs to be able to distinguish between different types of elements, such as buttons, input fields, and drop-down menus,

and understand the associated actions. We develop a series of queries accordingly in Appendix B.1, and randomly select UI elements and the layout to assemble queries for the UI Agent.

(2) **Simple Instruction Knowledge**: The agents are tasked with performing basic interactions, such as clicking or typing, in response to simple instructions. Specifically, given the complete action space, we prompt the UIAgent to generate possible instructions related to a random UI element, and their corresponding responses. For example, in Figure 1, the UIAgent was prompted to generate an instruction for element 9 ("*selecting the M4a format*") and then generates the corresponding response to interact with it. By learning this type of knowledge, the agent lays the foundation for completing a complex sequential decision-making process.

(3) **Process Preference Knowledge**: Real-world interactive tasks is quite difficult, and even the most advanced large language model, GPT-4, shows a low task completion rate (30%) in the mobile environment AndroidWorld (Rawles et al., 2024). Training a model solely on scarce successful trajectories still inevitably results in errors. Therefore, as illustrated below Figure 1, we introduce the adversarial agent against the UI agent, and the critic agent to score all actions, obtaining process preference data with step-level rewards. By learning from process preference data, the agent can better distinguish between correct and incorrect actions during the process, ultimately improving task completion rates. The distribution of the collected data can be found in Appendix B.2.

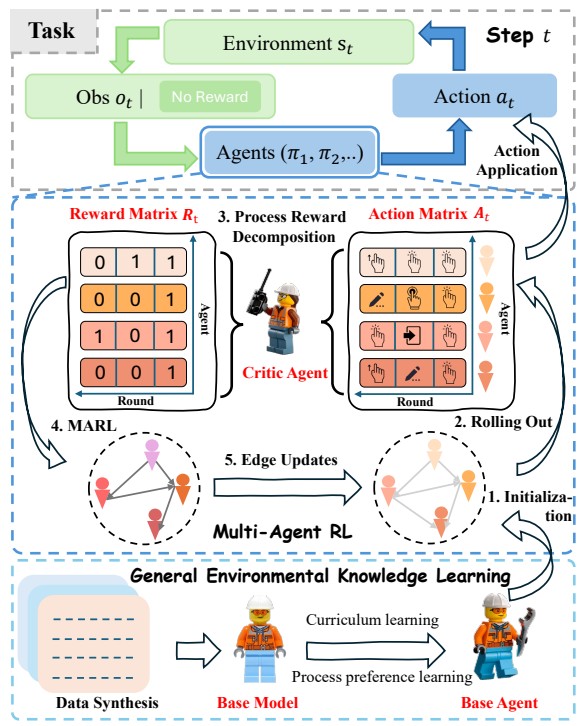

Figure 2: The multi-agent reinforcement learning stage based on process reward decomposition. Edge updates happen before rolling out. The Critic Agent at each step assess the scores of the whole action matrix to get the reward matrix and updating the agents accordingly.

The base model is first trained using Supervised Fine-Tuning (SFT) on the basic environmental knowledge and the simple instruction knowledge, progressively. The learning objective is:

$$\mathcal{L}_{\text{SFT}} = -\mathbb{E}_{(s,a)\sim\mathcal{D}}\left[\log \pi_\theta(a|s)\right], \quad (6)$$

where $\mathcal{D}$ represents the dataset of state-action pairs. Following SFT, the base model are further optimized using Direct Preference Optimization (DPO) on the process preference knowledge:

$$\mathcal{L}_{\text{DPO}} = -\mathbb{E}_{(s,a_+,a_-)\sim\mathcal{P}}\left[\log \sigma \left(\beta\log\frac{\pi_\theta(a_-|s)}{\pi_{\text{ref}}(a_-|s)} - \beta\log\frac{\pi_\theta(a_+|s)}{\pi_{\text{ref}}(a_+|s)}\right)\right], \quad (7)$$

where $\mathcal{P}$ is the preference-labeled dataset, $a_+, a_-$ denote positive and adversarial actions, $\sigma$ is the sigmoid function, $\beta$ is the hyper-parameter, and $\pi_\theta, \pi_{\text{ref}}$ are the base model and reference model (could be omitted for online optimization). For clarity, the DPO process could be either online, that keep updating the base model as the UI agent, or offline, that collect all the data at once.

### 2.2.3 STAGE 2: MULTI-AGENT REINFORCEMENT LEARNING

In the second stage of the *CollabUIAgents* framework, we address the challenge of sparse rewards in interactive dynamic environments by introducing a novel **Process Reward Decomposition** strategy for multi-agent reinforcement learning (MARL). This approach provides fine-grained reward signals at both the agent and conversation round levels, enabling agents to learn more effectively from their interactions and improve awareness towards multi-agent collaboration.

**Process Reward Decomposition** By expanding the critic agent that provides process rewards at each step to the multi-agent system, we further allocate rewards in a finer granularity, at both the

agent level and the conversation round level. The whole process is visualized in Figure 2. At each time step $t$, we collect the actions $a_t^i$ from all agents $\pi_i$ in the system $\mathcal{G}$, forming the **action matrix** $\boldsymbol{A}_t$ as described in Section 2.2.1. The critic agent assesses these actions based on the task and current environment state individually, generating a **reward matrix** that provides reward feedback for each agent's action at each conversation round:

$$\boldsymbol{R}_t = (r_t^{i,j}), i = 1, ..., n, j = 1, ..., m,, \tag{8}$$

where $r_t^{i,j}$ denotes the intermediate reward from agent $\pi_i$ at $j$-th conversation round for interaction step $t$, reflecting the quality or contribution of agent $\pi_i$'s action for task solving. The total reward for the task is then decomposed as:

$$R_{\text{total}} = \bigvee_{t=1}^{n} \bigvee_{i=1}^{n} \bigvee_{j=1}^{m} r_t^{i,j}, R, r_t^{i,j} \in \{0, 1\}. \tag{9}$$

For the circumstance that $R = 1$, $r_t^{i,j} = 1$ is guaranteed for at least one $t, i, j$. The rationale is that, for the critic agent, it might be more simple to identify whether a single decision is wrong, than to judge the reward of long decision chains between multiple agents. Thus, we hypothesize that by tearing down the granularity, the quality of the reward signal would not fall behind the end-to-end reward provided by the environment. Instead, this decomposition provides a more detailed reward signal, enabling agents to adjust their behavior based on individual contributions and collaborative success, even when the end-to-end reward is sparse. Qualitative study is shown in Appendix A.1.

**MARL with Edge Updates**    To optimize the agents' policies in this multi-agent setting, the overall objective is related to Value Decomposition Proximal Policy Optimization (VDPPO) (Ma & Luo, 2022), which is designed for cooperative multi-agent environments. Instead of setting up critics, we adopt DPO training with preference data synthesis similar to Section 2.2.2 for efficiency. Different from VDPPO settings, agents in the system could communicate and the message network should also be updated in the optimization. To alleviate the overhead of learning the optimal combination of edges, we introduce an *edge update* trick, that randomly update edges to form a DAG for message passing between agents. Through this process, we encourage agents to learn the awareness of multi-agent collaboration and adapt to diverse message networks rather than being rigid in locally optimal DAG pattern. As shown in Figure 2, the edge update is functioned before rolling out actions from the policy models. The overall learning objective for each agent $\pi_i$ is formulated as:

$$\mathcal{L}_{\text{MARL}}(\theta_i) = -\mathbb{E}_{(s_t, a_t^{i,+}, a_t^{i,-}) \sim \mathcal{P}(\mathcal{G}, \mathcal{E}_{\mathcal{G}}' \sim K_{|\mathcal{G}|})} \left[ \log \sigma \left( \beta \left( \log \pi_{\theta_i}(a_t^{i,+}|s_t) - \log \pi_{\theta_i}(a_t^{i,-}|s_t) \right) \right) \right], \tag{10}$$

where $\theta_i$ are the parameters of agent $\pi_i$, $K_{|\mathcal{G}|}$ is a fully connected graph of $|\mathcal{G}|$ nodes, $\mathcal{E}_{\mathcal{G}}'$ represents a DAG subgraph sampled from $K_{|\mathcal{G}|}$, and $\mathcal{P}(\mathcal{G}, \mathcal{E}_{\mathcal{G}}')$ is the preference dataset sampled with agents in the message network $\mathcal{E}_{\mathcal{G}}'$. This objective encourages the policy $\pi_{\theta_i}$ to assign higher probabilities to preferred actions $a_t^{i,+}$ compared to less rewarded actions $a_t^{i,-}$. The agents' policies could be updated online or offline as well, and, throughout the MARL process with edge updates, the edge connections in the communication graph $\mathcal{E}$ among agents can also be configured during inference time, allowing the system to adjust communication pathways for better collaboration.

## 2.3 CROSS-ENVIRONMENT ADAPTATION

One of the key strengths of the *CollabUIAgents* framework is its ability to generalize across different interactive environments, such as across mobile operations and web browsing environments. The framework supports two ways of adaptation.

**Direct Transfer**    In scenarios where the new environment shares similarities with the training environment, agents can be directly deployed without additional training. For example, agents trained in mobile UI environments can directly apply their knowledge to web environments, leveraging the knowledge common interaction patterns and UI elements. The multi-agent setup may also decrease error rates through collaborations for expectation.

**Continual MARL**    When the new environment presents significant differences or the higher success rates are demanded, agents can undergo further training using the MARL framework with Process Reward Decomposition in the new environment. This continual reinforcement learning allows agents to refine their policies and adapt to new action spaces, or observation structures.

Table 1: Success Rates (SR) in AndoridWorld and MobileMiniWoB++ (MMiniWoB++).

| System | Base model | #Params | #Agents | Input | $SR_{AndroidWorld}$ | $SR_{MMiniWoB++}$ |
|---|---|---|---|---|---|---|
| *Agents based on Closed-Source LLMs* | | | | | | |
| M3A | GPT-4 | N/A | 1 | Text | **30.6** | 59.7 |
| M3A | Gemini 1.5 Pro | N/A | 1 | Text | 19.4 | 57.4 |
| M3A | GPT-4 | N/A | 1 | Text & Image | 25.4 | **67.7** |
| M3A | Gemini 1.5 Pro | N/A | 1 | Text & Image | 22.8 | 40.3 |
| SeeAct | GPT-4 | N/A | 1 | Text & Image | 15.5 | 66.1 |
| *Agents based on Open-Source LLMs* | | | | | | |
| Qwen2 | Qwen2 | 7B | 1 | Text | 6.2 | 12.9 |
| SingleAgent | Qwen2 | 7B | 1 | Text | 18.9 | 48.4 |
| GroupAgents | Qwen2 | 7B | 4 | Text | 21.4 | 53.2 |
| CollabUIAgents$_{mobile}$ | Qwen2 | 7B | 4 | Text | **29.3** | **61.2** |

## 3 EXPERIMENT

### 3.1 EXPERIMENTAL SETTINGS

**Environments**    We conduct experiments in both mobile and web environments. For the mobile environments, we use AndroidWorld (Rawles et al., 2024) and MobileMiniWoB++ (Rawles et al., 2024): (1) **AndroidWorld** has 116 programmatic tasks across 20 real-world apps, such as Chrome, Markor, and Pro Expense. (2) **MobileMiniWoB++** is derived from MiniWoB++ (Shi et al., 2017), which is a web-based benchmark. MobileMiniWoB++ shares the same observation space as AndroidWorld and supports 92 tasks from MiniWoB++. We use the success rate (SR) as an evaluation metric. For the web environments, we leverage Mind2Web (Deng et al., 2023) and AutoWebBench (Lai et al., 2024): (1) **Mind2Web** features over 2,000 open-ended tasks sourced from 137 websites in 31 different domains. (2) **AutoWebBench** is a bilingual benchmark featuring approximately 10,000 traces, from mainstream Chinese and English websites, providing a diverse dataset for web browsing. We use the step-success rate (SSR) as the evaluation metric.

**Evaluated Methods**    We compare our framework against the following existing methods: (1) **M3A** (Rawles et al., 2023) is a multimodal autonomous agent, which combines ReAct-style (Yao et al., 2022b) and Reflexion-style (Shinn et al., 2024b) prompting to interpret user instructions and screen content, then reason and update its decision-making based on the outcome of its actions. (2) **SeeAct** (Zheng et al., 2024) is a navigation agent originally designed for GPT-4V to perform actions through textual choices. To adapt it to the Android environment, the action space was expanded to support mobile-specific actions. (3) **SeeClick** (Cheng et al., 2024) is a visual GUI agent that automates tasks by solely relying on screenshots. It employs GUI grounding to enable the agent to accurately locate interface elements based on user instructions. We leverage Qwen2 7B as our base model and evaluate the following systems derived from the model: (1) **SingleAgent** is the base model that has undergone the stage 1 in our framework. (2) **GroupAgents** is a direct combination of multiple single agents, which are interconnected by random edges forming a message network as described in Secion 2.2.1. They select actions through majority voting for a round. (3) **CollabUIAgents$_{mobile}$** is our method applied on AndroidWorld with $n = 4, m = 3$. (4) **CollabUIAgents$_{m \to web}$** builds upon CollabUIAgents$_{mobile}$ with continue MARL on the training set to adapt to Mind2Web. Due to computational resource limits, we adopted offline training for reinforcement learning in all methods.

### 3.2 MAIN RESULTS

**Effectiveness in Mobile Environments**    In this section, we explore the effectiveness of our proposed method for both in-domain tasks and cross-task generalization. Experimental results in mobile environments are shown in Table 1. The best performance is achieved by GPT-4 without additional training, consistent with findings from other studies indicating that closed-source LLMs like GPT-4 and Gemini 1.5 Pro are high-performing generalists. In contrast, the open-source LLM Qwen2 initially shows low performance in its vanilla form ("Qwen2" in Table 1). However, after fine-tuning with data from the AndroidWorld environment, its performance improves significantly, highlighting the effectiveness of the fine-tuning process. Moreover, notable performance gains are observed when multiple agents are utilized ("SingleAgent" vs. "GroupAgents"). Our proposed multi-agent

Table 2: Step Success Rates (SSR) in the Mind2Web environment. * indicates fine-tuning the model on the corresponding training set.

| System | #Params | #Agents | Input | Cross-Task | Cross-Website | Cross-Domain | Avg. |
|---|---|---|---|---|---|---|---|
| *Agents based on Closed-Source LLMs* | | | | | | | |
| GPT-3.5-Turbo | N/A | 1 | Text | 17.4 | 16.2 | 18.6 | 17.4 |
| GPT-4 | N/A | 1 | Text | **36.2** | **30.1** | **26.4** | **30.9** |
| *Agents based on Open-Source LLMs* | | | | | | | |
| Qwen-VL* | 9.6B | 1 | Text & Image | 12.6 | 10.1 | 8.0 | 10.2 |
| SeeClick* | 9.6B | 1 | Text & Image | 23.7 | 18.8 | 20.2 | 20.9 |
| Qwen2 | 7B | 1 | Text | 8.6 | 6.3 | 7.5 | 7.4 |
| SingleAgent | 7B | 1 | Text | 13.4 | 10.6 | 11.8 | 11.9 |
| GroupAgents | 7B | 4 | Text | 15.7 | 11.2 | 12.9 | 13.2 |
| CollabUIAgents$_{mobile}$ | 7B | 4 | Text | 19.2 | 13.8 | 15.5 | 16.2 |
| CollabUIAgents$_{m \rightarrow web}$ | 7B | 4 | Text | **34.5** | **32.7** | **25.1** | **30.7** |

Table 3: Step Success Rates (SSR) of different models in the AutoWebBench environment. All systems are evaluated with in-context learning prompts presented in Appendix C.

| System | #Params | #Agents | English | | Chinese | | Avg. |
|---|---|---|---|---|---|---|---|
| | | | Cross-Task | Cross-Domain | Cross-Task | Cross-Domain | |
| *Agents based on Closed-Source LLMs* | | | | | | | |
| GPT-3.5-Turbo | N/A | 1 | 12.1 | 6.4 | 13.5 | 10.8 | 10.7 |
| GPT-4 | N/A | 1 | **38.6** | **39.7** | **36.7** | **36.3** | **37.8** |
| Claude2 | N/A | 1 | 13.2 | 8.1 | 13.0 | 7.9 | 10.5 |
| *Agents based on Open-Source LLMs* | | | | | | | |
| LLaMA2 | 7B | 1 | 3.3 | 2.5 | - | - | 2.9 |
| LLaMA2 | 70B | 1 | 8.3 | 8.9 | - | - | 10.6 |
| Qwen2 | 7B | 1 | 8.6 | 9.4 | 8.1 | 7.8 | 8.5 |
| SingleAgent | 7B | 1 | 12.0 | 13.3 | 12.7 | 13.4 | 12.8 |
| GroupAgents | 7B | 4 | 13.7 | 14.5 | 15.0 | 13.9 | 14.0 |
| CollabUIAgents$_{mobile}$ | 7B | 4 | 18.6 | 17.7 | 19.1 | 15.6 | 17.7 |
| CollabUIAgents$_{m \rightarrow web}$ | 7B | 4 | **34.3** | **36.9** | **35.3** | **32.5** | **34.7** |

framework further enhances performance, achieving the best results among systems based on open-source LLMs ("CollabUIAgents$_{mobile}$"). Remarkably, it outperforms Gemini 1.5 Pro in both test environments and achieves performance comparable to or better than GPT-4. These outcomes demonstrate the effectiveness of our framework in dynamic environments. Additionally, even though our CollabUIAgents$_{mobile}$ has no prior exposure to evaluation tasks from the MobileMiniWoB++ environment, it still achieves substantial performance improvements on these tasks, demonstrating its strong generalization capability to out-of-domain tasks.

**Generalizing from Mobile to Web Environments** In this section, we examine the cross-environment generalization capabilities of our proposed method. Results for web environments are presented in Tables 2 and 3, corresponding to the Mind2Web and AutoWebBench environments, respectively. First, similar to the Android environments, vanilla Qwen2 ("Qwen2" in Tables 2 and 3) demonstrates low performance in web environments. In contrast, both fine-tuning ("SingleAgent") and multi-agent ("GroupAgents") approaches contribute to performance improvements, though the gains are relatively smaller compared to those observed in the Android environments. Second, applying the agent system obtained from the AndroidWorld environment using our proposed method to the web environments ("CollabUIAgents$_{mobile}$") yields performance improvements; however, these absolute gains remain modest. This suggests that while our method exhibits some cross-environment generalization ability, there is still considerable room for enhancement. Third, we continue to fine-tune MA-Android using MARL on data collected from Mind2Web, leveraging our multi-agent data synthesis pipeline. As shown in Table 2 ("CollabUIAgents$_{m \rightarrow web}$"), this results in substantial performance gains, achieving results comparable to GPT-4. It is noteworthy that we do not require human-annotated data for the Mind2Web environment, which is a significant advantage in transferring the agent system to new environments. Finally, results in Table 3 ("CollabUIAgents$_{m \rightarrow web}$") indicate that the agent system obtained from the Mind2Web environment using our method generalizes well to the AutoWebBench environment, achieving results comparable to GPT-4. This demon-

Table 4: Ablation study. Success Rates (SR) in the AndroidWorld and MobileMiniWoB++ (MMini-WoB++) environments are reported.

| System | #Params | #Agents | $SR_{AndroidWorld}$ | $SR_{MMiniWoB++}$ |
|---|---|---|---|---|
| *Stage 1* | | | | |
| Qwen2 | 7B | 1 | 6.2 | 12.9 |
| + Basic knowledge SFT | 7B | 1 | 12.1 | 22.5 |
| + Instruction SFT | 7B | 1 | 15.1 | 35.8 |
| + Process DPO | 7B | 1 | **18.9** | **48.4** |
| *Stage 2* | | | | |
| GroupAgents w/ Vanilla Qwen2 | 7B | 4 | 8.6 | 16.1 |
| GroupAgents w/ Stage-1 Qwen2 | 7B | 4 | 21.4 | 53.2 |
| CollabUIAgents$_{mobile}$ | 7B | 4 | **29.3** | **61.2** |
| w/ MARL $\rightarrow$ MA-SFT | 7B | 4 | 23.2 | 54.8 |
| w/o reward decomposition | 7B | 4 | 25.0 | 56.4 |
| w/o edge update | 7B | 4 | 27.6 | 58.1 |
| CollabUIAgents$_{m \rightarrow web}$ | 7B | 4 | 26.7 | 58.1 |

strates the strong generalization capability of our method across tasks, consistent with observations in the Android environments.

## 3.3 ABLATION STUDY

The results of the ablation study are presented in Table 4. We conduct automated data synthesis, model training and evaluation in the AndroidWorld environment. Additionally, we directly apply the resulting system to the MobileMiniWoB++ environment for evaluation.

**Stage 1: Environment Adaptation** In this stage, we develop an automated data synthesis method to gather basic environmental knowledge, simple instruction knowledge, and process preference knowledge from the dynamic mobile environment, AndroidWorld. Based on the upper section of Table 4, we derive the following conclusions: (1) Incorporating basic environmental knowledge data substantially improves the base model's comprehension of dynamic mobile environments, achieving a absolute performance gain of 5.9% in AndroidWorld and 9.6% in MobileMiniWoB++ ("+ Basic knowledge SFT"). It is noteworthy that the collected UI page information excludes app-specific details of MobileMiniWoB++, yet training with general knowledge from AndroidWorld enables the model to generalize effectively to new apps and tasks. (2) Simple instruction knowledge data is crafted to guide the agent in interacting with the environment using actions from the specified action space. Our experiments demonstrate that incorporating instruction data further enhances the base model's ability to complete simple tasks within UI environments ("+ Instruction SFT"). (3) A key advantage of our proposed method is its ability to learn from incorrect actions using process preference knowledge data. Experimental results confirm that this addition significantly boosts performance ("+ Process DPO"). The improvement is more pronounced in the MobileMiniWoB++ environment, which we attribute to the simplicity of its tasks. Fewer steps are required to complete these tasks, leading to greater performance gains.

**Stage 2: Multi-agent Learning** This stage focuses on training multiple agents to collaborate and achieve superior results. The experimental findings, presented in the lower section of Table 4, highlight the following key insights: (1) Combining multiple agents based on a vanilla base model using random edges leads to modest improvements ("GroupAgents w/ Vanilla Qwen2" in Table 4). In contrast, substituting these agents with enhanced versions from stage 1 ("GroupAgents w/ Stage-1 Qwen2") results in significant performance gains, underscoring the importance of first enhancing individual agents before integrating them. (2) Further training of the GroupAgents with trajectory data using either SFT ("CollabUIAgents$_{mobile}$ w/ MARL $\rightarrow$ MA-SFT") or DPO ("CollabUIAgents$_{mobile}$ w/o reward decomposition") improves performance, with DPO showing superior results. The primary distinction between these methods is that SFT can only learn from correct actions, while DPO can learn from both correct and incorrect actions. Consequently, DPO is able to leverage a greater quantity and diversity of data, leading to marginal improvement. (3) Our proposed method ("CollabUIAgents$_{mobile}$") introduces process reward decomposition, providing more granular feedback that facilitates exploration of the large action space at each step. This accelerates the adaptation of the agent group to the environment, yielding the best overall results. (4) A comparison

between systems with and without edge optimization ("CollabUIAgents$_{mobile}$" vs. "w/o edge update") demonstrates that edge optimization contributes to further performance improvements. (5) After cross-environment reinforcement learning on the web, "CollabUIAgents$_{m \rightarrow web}$" exhibits impressive autonomous adaptability in the new environment, with only minor performance fluctuations in the original mobile environment, thereby validating the stability of our method.

## 4 RELATED WORK

**Agents on Interactive Environments**    Before the advent of today's foundation models, the development of agents capable of interacting with user interfaces relied on traditional RL and behavioral cloning. These methods were primarily used to simulate interactions such as mouse clicks and typing via the keyboard (Liu et al., 2018b; Li et al., 2020; Humphreys et al., 2022). However, recent advancements have shifted towards leveraging pre-trained foundation models. By applying in-context learning and fine-tuning techniques, these models are now employed across various platforms, including mobile interfaces (Yan et al., 2023; Wang et al., 2023; Hong et al., 2024; Rawles et al., 2023), web environments (Zhou et al., 2024a; Lai et al., 2024; Koh et al., 2024; Cheng et al., 2024; Deng et al., 2023), and desktop operating systems (Xu et al., 2024; Wu et al., 2024; Xie et al., 2024; Zhang et al., 2024a). Recently, there are emerging methods (Shinn et al., 2024a; He et al., 2024; Pan et al., 2024) designing process rewards for single-agent learning for better performance.

**Prompt-based Multi-agent Learning**    In recent years, collaboration among multiple LLM agents has proven effective for various tasks (Ning et al., 2023; Hao et al., 2023; Jiang et al., 2023). Recent studies have developed different interaction architectures and assigned agents in static patterns (Hong et al., 2023; Wu et al., 2023; Qian et al., 2024). However, employing a static architecture without team optimization may restrict the performance and generalization of LLM-powered agent. Chen et al. (2023b) selects a fixed number of agents from a set of manual prompt candidates via an additional LLM during each round of discussion. Zhuge et al. (2024b) unify language agent systems by describing them as optimizable computational graphs and develop optimization methods for nodes and edges, enabling automatic improvements of agent prompts and inter-agent orchestration. Liu et al. (2023) employ a feed-forward network to formulate the process of LLM-agent collaboration for arbitrary tasks and introduce an unsupervised algorithm to optimize the team of agents by the individual contributions of agent.

**Interactive Environments for Agents**    To effectively evaluate autonomous agents, it is essential to create environments that not only replicate real-world conditions but also deliver immediate reward signals when tasks are successfully completed (Abramson et al., 2022; Ruan et al., 2023; Rawles et al., 2023; Deng et al., 2023). MiniWoB++ (Shi et al., 2017) is a lightweight framework that features small, synthetic HTML pages with parameterized tasks, allowing for virtually unlimited task variability. For more specialized environments, WebShop (Yao et al., 2022a) simulates an e-commerce platform, offering scenarios akin to online shopping. WebArena (Zhou et al., 2024a) and its visual counterpart, VisualWebArena (Koh et al., 2024), simulate websites spanning up to four distinct domains, while WorkArena (Drouin et al., 2024) focuses on enterprise software with a set of 29 tasks designed for workplace settings. For desktop operating systems, OSWorld (Xie et al., 2024) provides both a user interface and programmatically generated rewards across nine different apps. GAIA (Mialon et al., 2024), on the other hand, is to assess an agent's proficiency in daily assistance. AndroidWorld (Rawles et al., 2024) improves upon OSWorld's method by dynamically generating starting states and introducing limitless variability in task objectives.

## 5 CONCLUSION

In this paper, we introduce CollabUIAgents, a two-stage multi-agent learning framework to address reward scarcity problems and aims at generalization across tasks and even environments. In the first stage, we propose a fully automated data synthesis that allows agents to go through curriculum learning on three-level general environmental knowledge, without human intervention. In the second stage, we propose a process reward decomposition strategy in MARL to assign rewards at both the agent and conversation round levels. Experimental results demonstrate that our framework effectively improves the environment adaptability of open-source language models, and achieves GPT-4-comparable multi-agent systems across mobile and web environments.

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

## A ENVIRONMENT

### A.1 EXAMPLE OF DYNAMIC UI INTERACTION

Figure 3 is an example of task execution steps in the AndroidWorld environment, where "action_type" represents the action taken, and "index" represents the index of the UI element. We have marked the positions of the relevant elements on the UI interface.

### A.2 ACTION SPACE IN ENVIRONMENTS

Tables 5 and 6 show the action spaces of agents in mobile and web environments, respectively.

Table 5: Action space in mobile environment.

| Action | Description |
|---|---|
| CLICK | Tap once on the element |
| DOUBLE_TAP | Quickly tap the element twice |
| SCROLL | Slide the screen to view more content |
| SWIPE | Quick swipe across the screen |
| INPUT_TEXT | Type text into the element |
| NAVIGATE_HOME | Return to the home screen |
| NAVIGATE_BACK | Go back to the previous screen |
| KEYBOARD_ENTER | Press the enter key |
| OPEN_APP | Launch an app |
| STATUS | Check task status |
| WAIT | Pause briefly |
| LONG_PRESS | Tap and hold on the element |
| ANSWER | Give a response |
| UNKNOWN | Undefined action |

Table 6: Action space in web environment.

| Action | Description |
|---|---|
| CLICK | Click at an element |
| HOVER | Hover on an element |
| SELECT | Select option in an element |
| TYPE_STRING | Type to an element |
| SCROLL_PAGE | Scroll up or down of the page |
| GO | Go forward or backward of the page |
| JUMP_TO | Jump to URL |
| SWITCH_TAB | Switch to i-th tab |
| USER_INPUT | Notify user to interact |
| FINISH | Stop with answer |

## B DATA COLLECTION DETAILS

### B.1 QUESTIONS LIST

The questions used for UI basic environmental knowledge generation are shown in Table 7.

### B.2 DETAILS OF THE COLLECTED DATA

The distribution of the collected data is shown in Table 8.

### B.3 PROMPTS FOR DIFFERENT AGENTS

Prompts for different agents are shown in Figures 4 to 10.

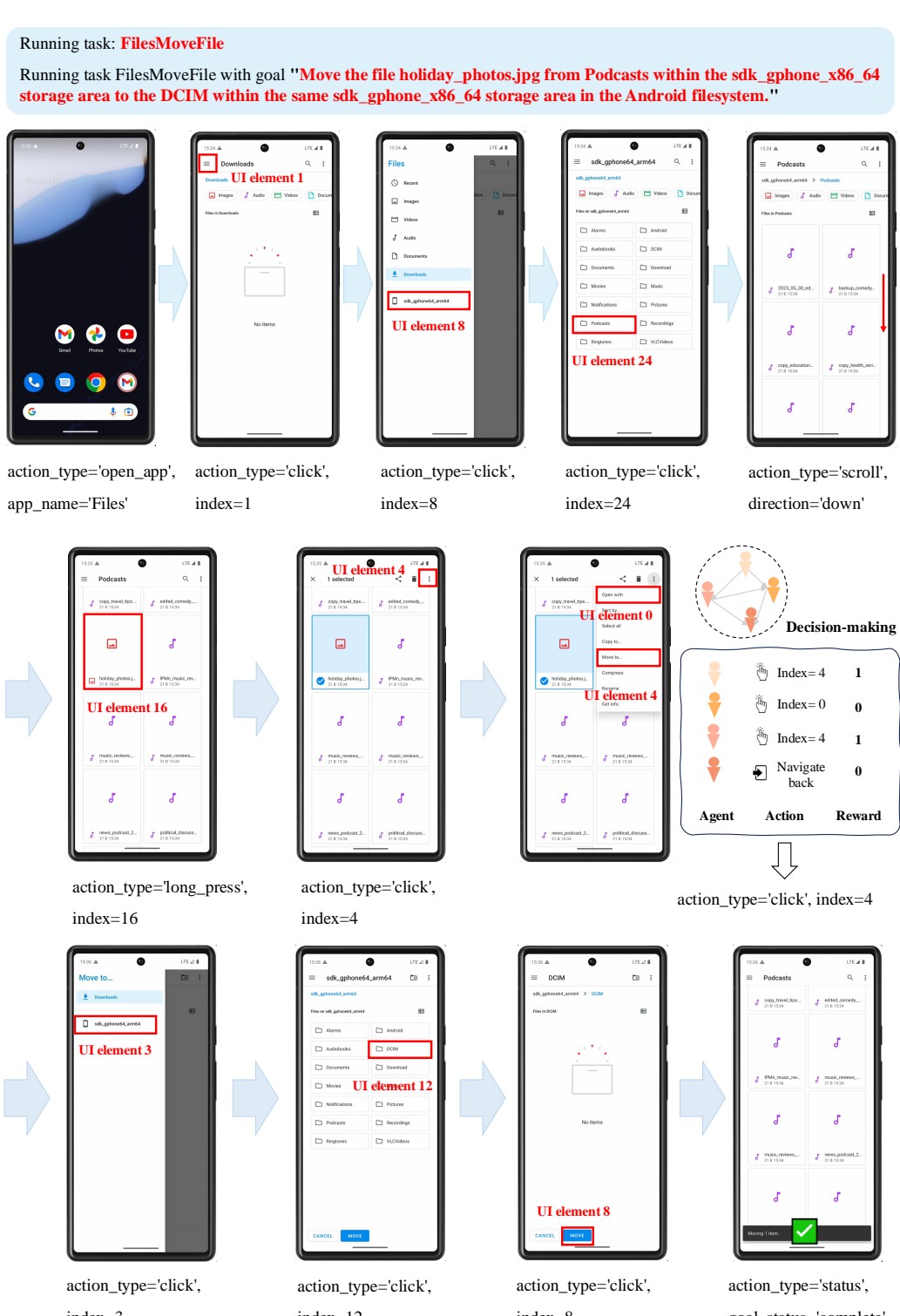

Figure 3: An example of task execution steps.

Table 7: Questions for UI basic environmental knowledge generation.

| Type | Question |
|---|---|
| UI Understanding | What is the purpose of the current UI? |
|  | What does the current UI aim to achieve? |
|  | Summarize the current interface in one paragraph. |
| Element Recognition | What is the function of UI element X? |
|  | What information does UI element X provide? |
|  | What happens when click the UI element X? |
|  | What action is associated with UI element X? |

Table 8: Collected data distribution.

| Data Type | Number |
|---|---|
| Basic Environmental Data | 88,513 |
| Simple Instruction Data | 18,041 |
| Process Preference Data | 3,440 |

The current user goal/request is: {goal}

Here is a history of what you have done so far: {history}

Here is a list of descriptions for some UI elements on the current screen:

{ui_elements_description}

General Guidance: {general_guidance}

Now output an action from the above list in the correct JSON format following the reason

why you do that. Your answer should look like:

'Reason: ...Action: {{"action_type":...}}'

Your answer:

Figure 4: The action prompt template for the UI agent.

You are an agent who can operate an Android phone on behalf of a user.

Here is a list of descriptions for some UI elements on the current screen:

{ui_elements_description}

Please answer the following questions for all the UI elements above.

Questions = ('
'What is the purpose of the current UI?'
'Summarize the current interface in one paragraph.'
'What does the current UI aim to achieve?
)

Please format your response as follows:
'{{"Question": "What is the purpose of the current UI?", "Answer":"........"}}'
'{{"Question": "Summarize the current interface in one paragraph.", "Answer":"........"}}'
'{{"Question": "What does the current UI aim to achieve?", "Answer":"........"}}'

Your response:

Figure 5: The UI understanding prompt template for the UI agent.

You are an agent who can operate an Android phone on behalf of a user.

Here is a list of descriptions for some UI elements on the current screen:

{ui_elements_description}

Please answer the following questions for all the UI elements above.

Questions = (
'What is the function of UI element X ?'
'What information does UI element X provide ?'
'What happens when click the UI element ?'
'What action is associated with UI element X ?'
)

Please format your response as follows:
'{{"Question": "What is the function of UI element X?", "Answer":"........"}}'
'{{"Question": "What information does UI element X provide?", "Answer":"........"}}'
'{{"Question": "What happens when click the UI element X?", "Answer":"........"}}'
'{{"Question": "What action is associated with UI element X?", "Answer":"........"}}'

Your response:

Figure 6: The element recognition prompt template for the UI agent.

You are an agent who can operate an Android phone on behalf of a user.

Here is a list of descriptions for some UI elements on the current screen:

{ui_elements_description}

The action space of the agent: {action_space}

General guidance: {general_guidance}

Please propose diverse simple instructions (one-step tasks) as many as possible based on the agent\'s action space and the current UI elements above in the following format: (contains at least one but no more than two \'complete\' actions and no more than one \'answer\' action)'
'{{"Instruction": "......", "Response": "Reason: ... Action: {{"action_type":...}}"}}'

For example:
'{{"Instruction": "I need to start recording audio", "Response": "Reason: The recording settings are all configured, I need to click \'Apply\' to apply the current settings and start recording. Action: {{"action_type": "click", "index": 3}}"}}'
'{{"Instruction": "I want to select the M4a format for recording.", "Response": "Reason: The recording format has been set correctly. Action: {{"action_type": "status", "goal_status": "complete"}}"}}'

'Your response:

Figure 7: The instruction generation prompt template for the UI agent.

The current user goal/request is: {goal}

Here is a history of what you have done so far: {history}

Here is a list of descriptions for some UI elements on the current screen:

{ui_elements_description}

General guidance: {general_guidance}

Now you need to role-play a very clumsy agent that can only output incorrect answer (if you have no choice, you can make up a wrong action and reason) from the above list in the correct JSON format, following the reason why you do that.

Your answer should look like:
'Reason: ...Action: {{"action_type":...}}'

Your answer:

Figure 8: The prompt template for the adversarial agent.

## C  PROMPT FOR AGENT WEB BROWSING

Prompts for agent web browsing is shown in Figures 11.

You are a super-intelligent agent who can expertly operate an Android phone on behalf of a user.

Now, you need to act as a critic, evaluating the actions taken by other Android agents.

These agents receive user tasks and current Android interface information and then take the next step.

Your evaluation should be between [0,1]. A score close to 0 means the agent's action is useless or incorrect in achieving the user's task, a score close to 0.5 means you are uncertain whether the agent's decision is useful for achieving the user's task, and a score close to 1 means the agent's action is useful or correct in achieving the user's task.

The current user goal/request is: {goal}

Here is a history of what have done so far: {history}

Here is a list of descriptions for some UI elements on the current screen:

{ui_elements_description}

General guidance: {general_guidance}

Here are the next actions different agents would like to take: {agents_actions}

Please output each agent's score in the correct JSON format, following the reason why you think the agents' actions and reasons are correct or not, and ensuring that their actions are necessary and not redundant for achieving the user's goals when you give your scores.

Figure 9: The prompt template for the critic agent.

Your answer should look like:

'Reason: The goal is {{user goal}}...Score: {{"agent_id": score...}}'

Here are some demonstrations of evaluations:\n'

1. Reason: The goal is ... Agent 0 and Agent 2 attempt to scroll down to find additional options. This is a logical step given that no explicit save button is visible and the app might have additional options accessible through scrolling. Agent 1 decides to click the "Settings" button in hopes that it might lead to a menu with a save option. However, this seems less directly connected to saving the recording as the Settings menu is generally for configuration rather than saving recordings. Score: {{"agent_0": 0.9, "agent_1": 0.1, "agent_2": 0.9}}.

2. Reason: The goal is ... All agents (Agent 0, 1, and 2) have chosen to input the desired name "xxx.m4a" into the text field, However, user did not specify a name. This is the incorrect next step, ...Score: {{"agent_0": 0.2, "agent_1": 0.2, "agent_2": 0.2}}.

3. Reason: The goal is ... Historical information shows that the agent has taken the same action multiple times. I am unsure if taking the same action again is reasonable. Score: {{"agent_0": 0.5, "agent_1": 0.5, "agent_2": 0.5}}.

Now output each agent's score.

Your answer must in the format:

'Reason: The goal is {{user goal}}...Score: {{"agent_id": score, "agent_id": score, "agent_id": score}}'

Your Evaluation:

Figure 10: The few-shot prompt template for the critic agent.

<html > {html_content} </html >

You are a helpful assistant that can assist with web navigation tasks. You are given a simplified html webpage and a task description. Your goal is to complete the task. You can use the provided functions below to interact with the current webpage.

#Provided functions: {action_space}

#Previous commands: {previous_commands}

#Window tabs: {exist_window_tabs_with_pointer_to_current_tab}

#Current viewport (pages): {current_position} / {max_size}

#Task: {task_description}

You should output one command to interact to the currrent webpage. You should add a brief comment to your command to explain your reasoning and thinking process.

Figure 11: The input prompt template for agent web browsing.

