# OpenReview forum: "Enhancing Multi-Agent Learning in Real-World Interactive Environments through Process Reward Decomposition"
_ICLR.cc/2025/Conference — Submitted to ICLR 2025_

### Official Review · Reviewer_hJ5t · 2024-10-27

**Soundness:** 2
**Presentation:** 2
**Contribution:** 2
**Rating:** 3
**Confidence:** 4

**Summary:**

This paper introduces a two-stage multi-agent learning framework to address the poor generalization and sparse reward issues in large language model (LLM)-based agent systems. In the first stage, the authors propose a data synthesis pipeline to automatically generate training data for fine-tuning the LLM. In the second stage, they leverage a critic agent to allocate rewards to each agent at every conversation round.

**Strengths:**

1. This paper studies an important problem: the generalization of cross-platform.
2. The idea of assessing the contributions of each agent is interesting.

**Weaknesses:**

1. The organization of the paper needs some improvement, especially MARL with Edge Updates Section.
2. The main concern is that the proposed Process Reward Decomposition is not well-justified: I am skeptical about using a critic agent to directly generate temporal and structural credit assignments.
3. Lack of comparison with related work on multi-agent LLM[a].


a. Zhuge, Mingchen, et al. "Language agents as optimizable graphs." arXiv preprint arXiv:2402.16823 (2024).

**Questions:**

1. Equation 9 needs more explanation.
2. Not strongly connected to MARL.

---

### Official Review · Reviewer_Uinp · 2024-11-01

**Soundness:** 2
**Presentation:** 2
**Contribution:** 2
**Rating:** 3
**Confidence:** 4

**Summary:**

This paper addresses the deficiencies in flexibility and generalization in existing multi-agent LLMs and proposes the CollabUIAgents framework based on general environment knowledge learning and MARL in two stages. This enhances performance in Open-Source LLMs. However, the paper still needs improvement in specific expression, motivation, and writing style, as detailed below.

**Strengths:**

This paper investigates an important direction in multiagent systems and poses a promising direction for future research.

**Weaknesses:**

1. In Task Formulation, it is necessary to clarify the concepts of  `agent` and `policy`. Furthermore, there needs to be consistency between a_t in Eq(3) and Eq(1). It is unclear whether a_t represents an action by a single agent, a joint action by multiagent, or an aggregation action, which should be specified.

2. In stage 1, why is the curriculum divided into three parts? or what is the insight?  It's better to explain the rationale behind choosing these three particular parts for the curriculum, and how each part contributes to the overall learning process.

3. How to comprehend  ''The rationale is that, for the critic agent, it might be more simple to identify whether a single decision is wrong, than to judge the reward of long decision chains between multiple agents''? It seems to contradict RL research. Instead, single-step rewards are usually not as accurate as long-term rewards, From the internal logic, judging a single-step decision is more difficult than multi-step.  It's better to provide evidence or reasoning to support their claim, especially in light of existing RL literature that suggests otherwise.

4. Multi-agent reinforcement learning methods are usually based on mathematical models such as Markov Game and Dec-POMDP. In stage 2, what is the multi-agent model, which needs a detailed explanation? How does it relate to or differ from standard models like Markov Games or Dec-POMDPs?

5. The nature of their reward decomposition process in stage 2  should be explained clearly. Is it artificial prior or adaptive learning? If it is artificial prior, it is no different from role allocation in form, and it is difficult to reflect the advantage of adaptive learning. How does it differ from or improve upon traditional role allocation methods?

6. In stage 2, it seems that each agent learns independently. How to ensure multi-agent collaboration? What are the mechanisms or techniques to promote collaboration between agents during the learning process in stage 2?

7. The paper also has several unclear character definitions and grammatical problems, such as：

    (1) It should be "each agent \pi_i” instead of  "all agents \pi_i” in line 271

    (2) It should be "the number of agents” instead of all "the number agents" in line 132

    (3) In Task Formulation, transition function and the maximum step are both represented by the character `T’.

    The reviewer suggests that the authors conduct a thorough proofreading of the paper, paying particular attention to consistency in mathematical notation and grammatical correctness.

**Questions:**

Please see the cons part.

---

### Official Review · Reviewer_G4vo · 2024-11-02

**Soundness:** 2
**Presentation:** 2
**Contribution:** 1
**Rating:** 3
**Confidence:** 2

**Summary:**

This paper investigated how to improve the performance of LLMs structured in multi-agent systems in interactive tasks. To this end, it proposed a two-stage strategy: (1) Synthesizing data automatically and training the LLM agents with curriculum learning based on the data; (2) generating reward decomposition across both decision making steps and negotiation step among agents at each decision making step. The proposed approach has been shown to enjoy substantial performance improvement against open-source models and a comparable performance to closed-source models.

**Strengths:**

1. It is difficult to evaluate the originality of this paper, as it mainly combines multiple existing techniques together to form a strategy for a specific application on LLMs. From the perspective of software engineering and empirical study, this paper lies in the category of original work, but the novelty is limited. It only addressed some evident weaknesses with unsurprising empirical strategies. However, some hypotheses seems insightful, such as the hypothesis to explain the randomly generated edges on communicaton graphs.
2. As an empirical study paper, it focused on proposing some hypothesis and methodology to address a problem, which have been well validated by the experimental results with ablation study to emphasize the importance of each module proposed. For this reason, the overall quality of this paper is good.
3. The motivation of this paper has been well clarified, and the methodology description and experimental setups have been well stated. The experimental analysis is comprehensive and reasonable to justify the importance of the proposed approach.
4. The significance of this paper is not easy to evaluate. Standing from the view of LLM performance improvement, the strategies revealed can make some contribution to the research field (but still waiting for reproducing the results). From the perspective of methodology, I cannot see any novel ideas. Overall, I believe this paper may only attract attention in the domain of LLMs, but I seriously suspect if the result of this paper can give a long-standing impact to the mainstream of ML.

**Weaknesses:**

1. In the majority voting described in equation (5), how do you process the situation where two or more actions are tied to have the same counts? If the strategy is random selection, could you please show me if this kind of strategy will incur some fluctuation in performance?
2. In line 171-172, the sentence that "The proper size of local memory enhances the diversity of decision making and avoids introducing too long contexts," is not easy to comprehend. I cannot have a clear picture to link the local memory size with the diversity of decision makings. Could you give more explanation about it?
3. I scanned the paper, and have not noticed any discussion on the faithfulness of automatically generated data to the realistic situations. Without this guarantee, I cannot foresee the benefit of automatic generation of data for training LLMs, as the resulting harm to the society could be more severe than the improvement measured in quantitative metrics and paid human labours. How do you guarantee that the generated data is faithful to realistic situations, with no hallucinations?
4. During processing preference knowledge, you proposed to use SFT followed by DPO, but with no reason to explain this strategy. Could you please give more insights into this strategy?

**Questions:**

Please address the concerns in weaknesses.

---

### Official Review · Reviewer_gWHo · 2024-11-04

**Soundness:** 3
**Presentation:** 3
**Contribution:** 2
**Rating:** 3
**Confidence:** 4

**Summary:**

This paper presents the CollabUIAgents framework, which aims to improve multi-agent learning capabilities in real-world interactive environments, particularly addressing issues related to sparse rewards. By employing a two-stage learning process—general environmental knowledge adaptation and multi-agent reinforcement learning with process reward decomposition—the framework enhances adaptability and cross-task generalization.

**Strengths:**

The method effectively tackles the challenges of sparse rewards and rigid role assignments in multi-agent learning. The two-stage learning process not only enhances adaptability across various tasks and environments but also allows for automated data generation, reducing the need for manual annotation. Furthermore, the comprehensive experimental design, including extensive comparisons with state-of-the-art models and thorough ablation studies, provides strong empirical support for the framework's effectiveness. The results indicate impressive performance improvements, demonstrating the framework's potential to advance multi-agent systems in real-world interactive scenarios.

**Weaknesses:**

The experimental setup in this paper lacks clarity, particularly regarding the configuration and evaluation of different agent setups, which impacts the transparency and reproducibility of the experiments. Additionally, the framework is demonstrated with a relatively small number of agents, raising scalability concerns. As the number of agents grows, managing an undirected communication graph may become computationally expensive, potentially affecting performance. Furthermore, some figures could benefit from clearer annotations and layout to better convey the framework’s structure and processes.

**Questions:**

1. **Scalability with Increased Agents**: The framework appears to use a relatively small number of agents. How does the method scale with larger numbers of agents, and what strategies could be implemented to address potential computational overhead in managing the communication graph?
2. **Edge Update Strategy in Graph Structure**: Could you clarify the role and frequency of edge updates within the communication graph? How does this edge updating process impact the overall performance, and would a static graph structure be a feasible alternative for certain applications?
3. **Experimental Parameter Choices**: Can you provide more context on the choices made for experimental parameters, such as the number of conversation rounds and agent configurations? How were these values determined, and could they impact the generalization of the results?
4. **Cross-Environment Transfer Learning**: While the framework shows promising results in cross-environment tasks, what specific techniques within the MARL setup contribute most to this adaptability? For instance, does the curriculum learning or specific reward design play a key role in enabling transfer learning?

---

### Meta-Review · Area_Chair_zUGs · 2024-12-19

**Metareview:**

The author did not respond during the rebuttal period, and the paper’s score was significantly below the acceptance threshold. As a result, the paper was decided to be rejected.

**Additional Comments On Reviewer Discussion:**

See above.

---

### Decision · Program_Chairs · 2025-01-22

Reject